# Obsidian forms by slow cooling

E. W. Llewellin [1], F. B. Wadsworth[2], P. Sullivan [1], J. P. Coumans [1], K. J. Dobson[3,4], M. C. S. Humphreys[1], A. Allabar[5], J. E. Gardner[6], R. A. Brooker [7], M. Nowak[5], T. Connolley [8], T. Havard [1,9] & C. Allgood [1,10]

Obsidian is a natural glass that is a common product of silicic volcanic eruptions. It has been prized throughout human history for its glassy nature, finding use in tools, weapons and ornaments, and in ritual and spiritual practice. The absence of large crystals in obsidian has led to the widespread view that it is formed by rapid cooling of lava, preventing crystal nucleation and growth. Here we show that, on the contrary, the absence of vesicles in obsidian requires relatively slow cooling – on the order of $10^{-4}$ to $10^{-8}$ °C/s – to enable the resorption of remnant bubbles. Our bubble-resorption model for obsidian formation is supported by in-situ X-ray computed tomography at magmatic temperatures that reveals bubble shrinkage during resorption on cooling. We validate a numerical model for growth and resorption of bubbles against these results, then apply the model to explore the conditions under which obsidian can form in nature. Our findings revise the accepted thermal histories of obsidian-forming systems, overturning conventional wisdom for the formation of this culturally, archaeologically, and volcanologically important material.

Obsidian is common in lavas produced by rhyolitic and phonolitic volcanic eruptions. It is also found in densely welded pyroclastic deposits, at the margins of shallow magmatic intrusions, and in blocks and pyroclasts produced by explosive eruptions[1]. In its type form it is highly glassy, containing no visible crystals or vesicles (Fig. 1), and is generally understood to represent the quenched molten-liquid component of the parent magma. The formation of obsidian is usually said to occur by rapid cooling of the lava edges, sometimes driven by contact with water or ice[2]. This mechanism is sufficiently ingrained that it has even entered popular culture: for example, in Minecraft™ – the best-selling video game of all time – lava turns immediately into obsidian on contact with water. Whilst it is true that the cooling rate must be sufficiently fast to avoid magma crystallisation during transition to the glassy state[3] this simple view misses a key challenge in the understanding of silicic eruptions: to explain why obsidian is typically vesicle-free and has no textural features that record the loss of volatile gases that exsolve from the magma during ascent.

At depth, silicic magmas have up to ~6 wt% dissolved volatile $H_2O$[4] whereas obsidian typically has <1 wt% dissolved $H_2O$[1,5], implying that most of the original $H_2O$ cargo has escaped during transport through the Earth's crust. Exsolved gas forms bubbles in magma[6], which are preserved as vesicles in solidified eruptive products. It is therefore enigmatic that most obsidian worldwide has <1 vol. % vesicles[1,7,8]. Two conceptual models have been proposed to resolve this conundrum: (1) the 'collapsing foam' model proposes that gas escapes through a permeable network of connected bubbles[8,9] and fractures[10]; (2) the 'cryptic fragmentation' model proposes that the ascending magma fragments explosively to form fine-grained ash particles that diffusively outgas $H_2O$ into a continuous gas phase before sintering to form a coherent melt in the shallow subsurface[5,11].

[1]Earth Sciences, Durham University, Durham, UK. [2]Earth and Environmental Sciences, Ludwig-Maximilians-Universität, Munich, Germany. [3]Department of Civil and Environmental Engineering, University of Strathclyde, Glasgow, UK. [4]Department of Chemical and Processes Engineering, University of Strathclyde, Glasgow, UK. [5]Department of Geosciences, University of Tübingen, Tübingen, Germany. [6]Jackson School of Geosciences, University of Texas at Austin, Austin, TX, USA. [7]School of Earth Sciences, Wills Memorial Building, University of Bristol, Bristol, UK. [8]Beamline I12, Diamond Light Source Ltd, Harwell Science and Innovation Campus, Didcot, UK. [9]Centrum Badań Kosmicznych Polskiej Akademii Nauk (CBK PAN), Warsaw, Poland. [10]Lancaster Environment Centre, Lancaster University, Lancaster, UK. ✉e-mail: ed.llewellin@durham.ac.uk

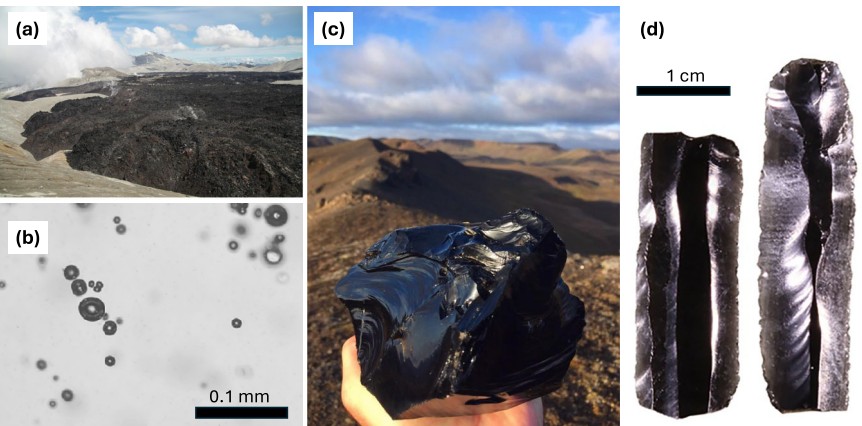

**Fig. 1 | Obsidian. a** Obsidian lava from 2011 eruption of Cordon Caulle (author photo). **b** Photomicrograph of section of obsidian pyroclast showing very low vesicle fraction (detail from ref. 39). **c** Obsidian block, Hrafntinnuhryggur, Krafla, Iceland (author photo). **d** Obsidian microblades from High-Arctic trade, 8000 BP (detail from ref. 40).

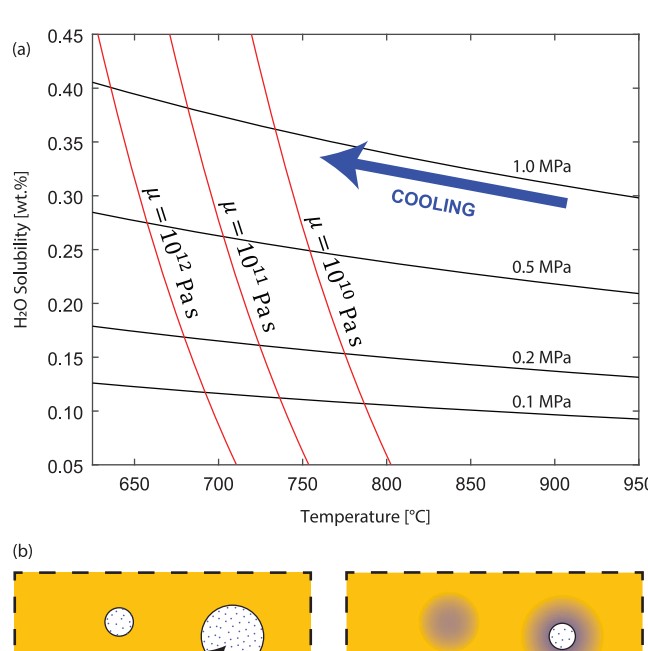

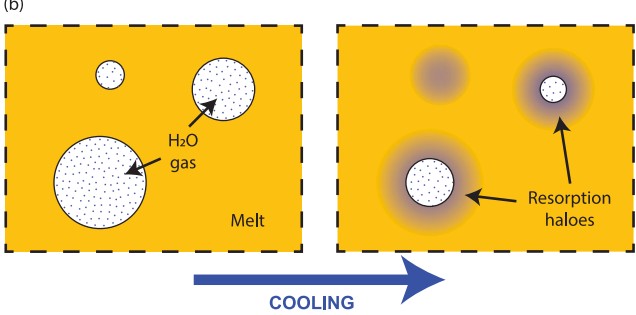

**Fig. 2 | Resorption of bubbles in silicate melt. a** Solubility of $H_2O$ in silicate melts increases for decreasing temperature. Black lines show solubility as a function of temperature at four different pressures for rhyolite composition. Red lines show three different isolines for viscosity $\mu(T)$, corresponding to the glass transition at different cooling rates. **b** Schematic showing resorption of $H_2O$ gas molecules (blue dots) into the melt surrounding bubbles (blue haloes) driven by increasing solubility during cooling[17].

Both models provide a mechanism by which most of the exsolved gas can escape from the magma. However, both also predict a non-zero final vesicularity $\phi$ that is trapped during densification as magma crosses the permeability 'percolation threshold' $\phi_c$. For the collapsing foam model, the percolation threshold occurs at $\phi_c \approx 30$ vol.%[12,13]; for the cryptic fragmentation model, this occurs at $\phi_c \approx 3$ vol.%[14]. Neither model can explain how the final 3–30 vol. % vesicularity that remains

after gas escape has completed, reduces to the <1 vol. % observed in dense obsidian.

Here we test the hypothesis that the final drop in vesicularity is not driven by percolating gas escape, but by a cooling-induced increase in $H_2O$ solubility in the magmatic melt, which creates the chemical potential to drive bubble resorption (Fig. 2). We focus on $H_2O$ because it is typically the most abundant volatile in silicic magmas.

As proof of concept, we perform an in-situ, high-temperature, synchrotron-source X-ray computed tomography experiment[15] using synthetic bubble-bearing obsidian. We subject the sample to controlled heating and cooling experiments to quantify the extent of bubble resorption. We use the standard synthetic obsidian composition 'AOQ8'[16] which we hydrate using a high-temperature sintering technique (see Methods). AOQ8 composition is given in the Supplementary Information. AOQ8 is used because it has a relatively high diffusivity and relatively low viscosity compared with natural rhyolite obsidian, hence thermally driven resorption can occur rapidly enough to observe on the timescale of a synchrotron experiment. The starting material for our in-situ experiment is a small cylinder of AOQ8 glass containing ~1 vol. % of small (~40 $\mu$m diameter) spherical vesicles containing argon and $H_2O$; the glass contains a dissolved $H_2O$ concentration at the solubility value (which is just below 0.1 wt%) for the synthesis conditions (1 atmosphere and 660 °C). We heat the sample in-situ, at 1 atmosphere, at a constant rate of 20 °C/min to 1065 °C, allow it to dwell isothermally at high temperature for 100 min, then cool at a constant rate of 20 °C/min back to ambient temperature. We collect 3D tomographs throughout the experiment, which we subsequently analyse to compute vesicularity (i.e., vesicle fraction expressed as a percentage). Vesicularity–time data are provided in the Supplementary Information.

## Results

The sample expands as bubbles grow from ~1 to ~16 vol. % vesicularity during heating and high-temperature dwell, then shrinks as bubbles resorb to ~4.5 vol. % during cooling (Fig. 3). Measurable expansion (and therefore bubble growth) begins when the temperature reaches around 700 °C, which is a little higher than the synthesis temperature. Expansion continues well into the isothermal dwell at 1065 °C, which is consistent with bubble growth that is limited by diffusion of $H_2O$ through the melt. Sample expansion on heating must be dominated by exsolution of $H_2O$ from the AOQ8 melt into the bubbles because equation-of-state expansion of the gas in the starting vesicles would, on its own, lead to a peak vesicularity of less than 2 vol. %. We attribute slow shrinkage of the sample over the second half of the dwell, during which vesicularity drops from ~16 to ~13 vol.%, to diffusively limited

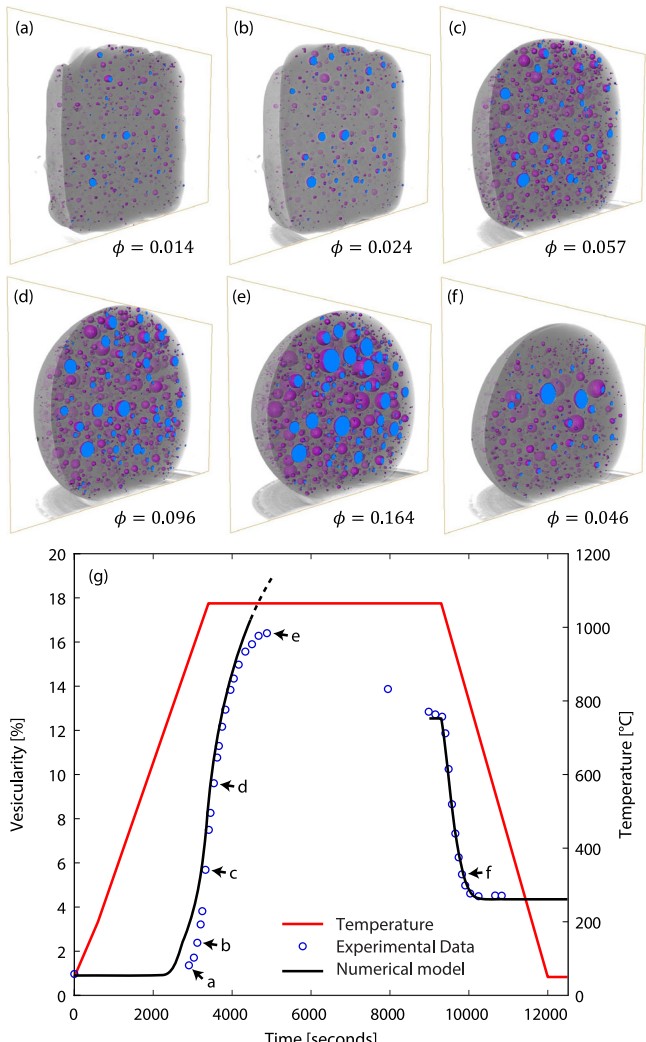

**Fig. 3 | Experimental observation of bubble resorption. a–f** Snapshots of 3D tomographs of sample during in-situ experiment; bubbles are rendered as purple spheres with blue interiors. Tomographs are keyed to plot shown in (**g**). **g** Experimentally observed evolution of vesicularity with time (blue circles, left axis) during heating and cooling (red line, right axis) of a sample of $H_2O$-bearing synthetic obsidian. Numerical model (black line, left axis) accurately captures experimental data trends. Mismatch (dashed line) between model and data during high-temperature dwell arises from diffusive loss of $H_2O$ from the bulk sample.

outgassing of the bulk sample into the dry atmosphere of the furnace. Vesicularity drops rapidly from 12.6 vol. % at the onset of the cooling ramp to 4.5 vol.% when the sample reaches around 750 °C, beyond which there is no further shrinkage of the sample. The shrinkage must involve substantial resorption of $H_2O$, because equation-of-state shrinkage would, on its own, only reduce vesicularity to around 10 vol. % between 1065 °C and 750 °C. Our data represent direct experimental observation of bubble shrinkage through thermally driven resorption – a process first hypothesized by McIntosh et al.[17].

We note that bubbles do not resorb to completion (i.e., bubble extinction) in our experiment because the glass transition is reached while diffusively limited resorption is ongoing, and because there is a small mass of relatively insoluble argon in the bubbles that was introduced deliberately during synthesis (see Methods).

The experimental data agree well with the predictions of a numerical model (Fig. 3) for the growth and resorption of bubbles in magma[18] (see Methods), except during the high temperature dwell, where diffusive outgassing from the sample (not captured by the model) is inferred to be important. This agreement extends the validation of the numerical model, previously validated only for bubble growth[18], to include resorption.

## Discussion

We now apply the validated numerical model to investigate natural scenarios in which bubbles can resorb to form dense obsidian (Fig. 4). We first explore the temporal evolution of vesicularity in cooling lava that has initial vesicularity $\phi_i = 30$ vol. % and bubble number density $N_b = 10^{14}$ m⁻³ (Fig. 4a) or $\phi_i = 3$ vol. % and $N_b = 10^{13}$ m⁻³ (Fig. 4b). Respectively, these scenarios represent cooling of the final products of the collapsing foam model (outgassed bubbly magma) and of the cryptic fragmentation model (sintered products outgassed at the point of fragmentation); appropriate bubble number densities for the two scenarios are taken from refs. 19 and 20. We assume an isobaric pressure $p = 1$ MPa, appropriate for the upper few metres of a lava. Resorption is modelled for different linear cooling rates $q$ from a typical eruption temperature for obsidian, $T_e = 825$ °C[21]. For the $\phi_i = 30$ vol. % case (Fig. 4a) we find that modest resorption occurs for $q \gtrsim 10^{-2}$ °C/s before the glass transition is reached, which takes a few hours, after which no further shrinkage can occur. However, for $q \lesssim 10^{-5}$ °C/s, bubbles fully resorb, to form dense obsidian, over around six months. When $\phi_i = 3$ vol. % (Fig. 4b) complete resorption occurs at $q \lesssim 10^{-3}$ °C/s, taking as little as a few days.

Next, we explore the effectiveness of resorption over a wide range of initial vesicularities and cooling rates (Fig. 4c) for the same conditions, taking $N_b = 10^{14}$ m⁻³. This analysis quantifies the 'window' in $\phi_i - q$ space in which dense obsidian can form by resorption. Results show that fully dense obsidian can form in a few days at $q \lesssim 10^{-3}$ °C/s for very low $\phi_i$ but slower cooling rates are required for higher initial vesicularities: for instance, lavas with $\phi_i \gtrsim 50$ vol. % require a few decades at $q \lesssim 10^{-7}$ °C/s to fully resorb. For fast cooling rates $q \gtrsim 10^{-2}$ °C/s, bubbles resorb only partially, and the final product remains vesicular after cooling.

Typical temperatures for eruptions that produce obsidian lie in the range $825 \leq T_e \leq 950$ °C[21,22] and obsidian-forming lavas have a typical half-thickness $5 \leq H \leq 30$ m[23]. Using these values, we can estimate the average cooling rate for natural obsidian-forming lavas by assuming conduction-limited cooling: $q_e \approx D_T T_e / H^2$, where $D_T \approx 10^{-6}$ m²/s is the thermal diffusivity for conduction-limited cooling[24]. This calculation yields cooling rates in the range $10^{-7} \lesssim q_e \lesssim 10^{-5}$ °C/s, which corresponds to cooling over a month to a decade, consistent with previous estimates[25]. This is sufficiently slow that lava with initial porosity up to around 50–60 vol. % can resorb to dense obsidian (Fig. 4) Plots similar to those in Fig. 4a-c, but for different initial conditions of $N_b$, $T_e$, and pressure, are presented in the Supplementary Information. Our study shows that dense obsidian is formed via a two-step process: (1) volatile-rich magma outgasses extensively during ascent, via either the collapsing foam or cryptic fragmentation mechanisms; (2) the residual vesicularity is lost through resorption during slow cooling. Heterogeneities in natural obsidian often take the form of flow bands with differing (low) vesicularity[26]; in our conceptual model these bands represent subtle differences in the initial vesicularity or initial $H_2O$ content of the bands prior to cooling.

The prevailing view that obsidian forms through rapid cooling stems from its crystal-poor nature. This observation must be reconciled with the slow-cooling requirement that we have demonstrated for bubble resorption. The time required for nucleation of crystals in $H_2O$-poor rhyolite, $\lambda_c$, was determined by Rusiecka et al.[27], who showed that their experimental results agreed well with nucleation theory[28]. They found that, under isothermal conditions, $\lambda_c$ is a strong function of temperature (Fig. 4d), with nucleation occurring most rapidly between 900 and 1000 °C. The minimum nucleation delay occurs at 950 °C, which gives $\lambda_c \approx 1.4 \times 10^9$ s (i.e., approximately 45 years).

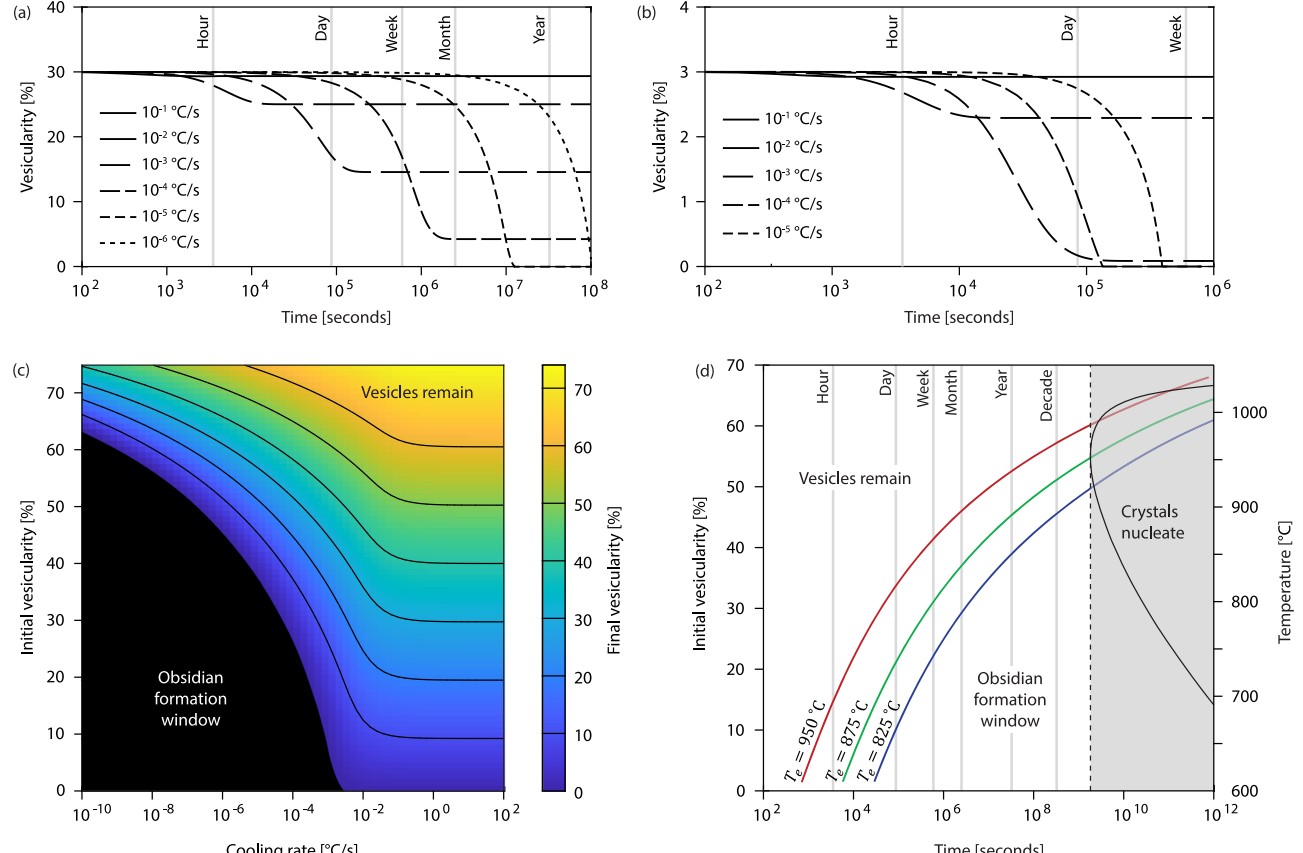

**Fig. 4 | Results of numerical modelling. a, b** Evolution of vesicularity over time for different cooling rates, for rhyolite samples starting from 30 and 3 vol. % vesicularity. **c** Map of initial and final vesicularity for rhyolite samples cooled at different rates. **d** Minimum time for complete bubble resorption for different initial vesicularity (left axis) and emplacement temperatures (red: $T_e$ = 950 °C; green: $T_e$ = 875 °C; blue: $T_e$ = 825 °C). Black line shows crystal nucleation delay $\lambda_c$[27] for different isothermal temperatures (right axis). Grey area shows window in which crystals may form.

We compare the minimum time for complete resorption of bubbles with the crystal nucleation time (Fig. 4d) for eruption temperatures $T_e$ = 825, 875, and 950 °C, bracketing the range of natural values[21,22], for $p$ = 1 MPa and $N_b$ = $10^{14}$ m$^{-3}$. This analysis reveals the critical window for obsidian formation – i.e., the region between the minimum time required to resorb bubbles, and the maximum time available before crystal nucleation may begin. We note that this is a deliberately conservative test—we take the minimum nucleation delay[27] for lava that remains at the optimal temperature for nucleation throughout, whereas any real cooling pathway necessarily involves the lava spending most time at sub-optimal temperatures for nucleation. This demonstrates that the 'rapid cooling' required to avoid crystallization is, in fact, not so rapid. We conclude that typical cooling rates for obsidian lavas are both rapid enough to avoid crystallization, and slow enough to allow bubble resorption.

Our model results and experiments show conclusively that bubbles resorb during cooling of magma, but that vesicles are preserved if the cooling is too rapid (faster than $10^{-4}$ to $10^{-8}$ °C/s, depending on starting bubble vol. fraction). No general mechanism has been proposed for the complete removal of bubbles from silicic lavas *before* cooling, so we conclude that obsidian formation requires slow cooling to resorb remnant bubbles. Since all lavas must cool, we propose that this provides a general and global mechanism for the formation of obsidian.

## Methods
### Sample synthesis
Crystal-free samples of nominally anhydrous standard synthetic peralkaline rhyolite were produced using the AOQ system $A_{38}O_{34}Q_{28}$,

where A, O, and Q refer to albite, orthoclase, and quartz, respectively. In all synthesis steps, iron-free crucibles were used. With respect to the AOQ system, we used a glass doped with 8 wt% Na$_2$O excess (termed AOQ8). Synthesis of this glass was achieved using high-grade reagents of SiO$_2$, Al$_2$O$_3$, Na$_2$CO$_3$, and K$_2$CO$_3$. The hydroscopic powders of SiO$_2$ and Al$_2$O$_3$ were first fired at 1073 K for 18 h, and the Na$_2$CO$_3$ and K$_2$CO$_3$ were dried at 383 K for 24 h. Then mixtures were made and blended in a shaker for 15 min, before being milled in a ball mill. The resultant powders were loaded in a Pt$_{90}$Rh$_{10}$ crucible, heated to 1273 K for 1 h, then to 1573 K for 3 h, and finally to 1873 K for 3 h. These steps ensured bubbles could escape the melt. The sample was then air-cooled. The glass was measured using electron microprobe analyses to check for homogeneity. The final glass was homogeneous and nominally CO$_2$-free. These glass samples were then crushed to around 50 μm particle size using a mortar and pestle and were formed into free-standing cylinders (3 mm diameter, 3 mm height) of powder held together by electrostatic forces[20] and loaded into a large quartz-glass tube in a horizontal furnace fitted with a nebulizer that flushed H$_2$O–argon mixed atmospheres past the samples at high temperature[29]. This step involved the diffusive hydration of the samples, and syn-hydration sintering to form a bubbly final melt with an elevated H$_2$O concentration. The composition of the gas was not monitored. Sintering synthesis was performed at 660 °C for 7 h yielding impermeable but bubbly glass with elevated concentrations of dissolved H$_2$O and argon-filled bubbles. The concentration of H$_2$O in the glass was confirmed using a Netzsch Simultaneous Thermal Analyser (heating up to 1400 °C) to be 0.09 ± 0.03 wt% H$_2$O. The use of argon as a carrier ensures that bubbles do not resorb to completion during post-sinter cooling. This is important for the subsequent beamline experiments

because bubbles are available to grow as soon as the glass transition temperature is exceeded, without the need for stochastic and poorly constrained nucleation of bubbles. Argon does not alter important material properties such as surface tension[30], and there is no evidence that this impacts viscosity or diffusivity of $H_2O$.

## Bubble growth and resorption experiments at the synchrotron-source beamline

We used the i12 beamline at the Diamond Light Source (EE16040) and a monochromatic 53 keV beam to capture 3D x-ray tomographs of the samples at high temperature. 1000 projections were collected per tomograph with 0.01 s exposure time. We used the PCO Edge camera with cropping to 80% of the field of view on Module 3 (3.25 $\mu m$ reconstructed voxel size), and a bespoke Severn Instruments clam-shell x-ray compatible single zone furnace (NERC-GeoX, University of Strathclyde) controlled using a Eurotherm programmable system. Data were reconstructed using standard Diamond i12 reconstruction pipelines[31–33]. Numerical modelling prior to the beamline experiments was used to design the heating/cooling profile to induce growth and shrinkage that is sufficiently rapid, and of sufficient magnitude, to be observed and recorded during reasonable experimental duration (a few hours), but sufficiently slow for clear imaging. Segmentation and analysis of the tomography was undertaken using Avizo™. Data were de-noised using an edge-preserving anisotropic diffusion filter, and segmented using a maximum entropy approach. The volume of the bubbles and melt phases was then quantified and used to calculate vesicularity.

## $H_2O$ solubility in magma

Liu et al.[34] compile published data for equilibrium concentrations of $H_2O$ (in wt%), $C_e$, in a rhyolitic magmatic melt from experiments performed at a range of $H_2O$ partial pressures $p_{H_2O}$ and temperature $T$. Using these data, they parameterize an empirical description of the form

$$C_e = \frac{a_1 p_{H_2O}^{1/2} + a_2 p_{H_2O} + a_3 p_{H_2O}^{3/2}}{T} + a_4 p_{H_2O}^{3/2} + \beta \tag{1}$$

where $a_1 = 354.94$, $a_2 = 9.623$, $a_3 = -1.5223$, and $a_4 = 0.0012439$ are fit coefficients of the model minimisation, $\beta$ is a term that captures the effect of dissolved magmatic $CO_2$ (neglected here) and, for these parameters, $p_{H_2O}$ and $T$ are in MPa and K, respectively. A similar approach was used to find the solubility of AOQ8[16], for which $a_1 = 536.4$, $a_2 = 5.125$, $a_3 = -1.091$, and $a_4 = 0.001323$.

## Magma bubble dynamics model for rhyolite magmas

We use a published magma bubble dynamics model[18] that is validated against bubble growth experiments using $H_2O$-bearing rhyolitic obsidian (from Krafla volcano, Iceland) at high temperature. The governing equation for the model is the Rayleigh–Plesset equation defined for a bubble of radius $R$ in a shell of melt of radius $S$ (both measured from the bubble centre) with a bubble gas pressure $p_g$, a shell liquid pressure $p_\infty$, a shell viscosity $\mu$, and a gas–melt interfacial tension $\Gamma$

$$p_g = p_\infty + \frac{2\Gamma}{R} + 12R^2 \frac{dR}{dt} \int_{R_i}^{S_i} \frac{\mu(x)x^2}{\left(R^3 - R_i^3 + x^3\right)^2} dx \tag{2}$$

where $R_i$ and $S_i$ are the initial values of $R$ and $S$, respectively, $t$ is time, and $x$ is Lagrangian radial coordinate. See Blower et al.[35] for the Lagrangian transformation of the bubble coordinate system, designed to simplify the integral in Eq. 2 by setting a coordinate system that does not depend on $R(t)$, and which is related to the Eulerian radial coordinate $A$ by $A^3 - R^3 = x^3 - R_i^3$. In Eq. 2, $\mu$ depends

on $x$ because $\mu$ is a function of the dissolved concentration of $H_2O$, $C$, which is in turn a function of $x$ via Fick's second law applied to the melt shell as follows

$$\frac{\partial C}{\partial t} = \frac{1}{x^2} \frac{\partial}{\partial x} \left( D \frac{A^4}{x^2} \frac{\partial C}{\partial x} \right) \tag{3a}$$

$$m = m_i + \frac{4\pi \rho_m}{100} \left( \int_{R_i}^{S_i} C(x,0)x^2 dx - \int_{R_i}^{S_i} C(x,t)x^2 dx \right) \tag{3b}$$

In Eq. 3, $D$ is the diffusivity of $H_2O$ in the melt, $m$ is the mass of gas in the bubble, $m_i$ is the initial mass, and $\rho_m$ is the melt density (taken here to be independent of $C$). The constitutive laws relating $\mu$ and $D$ to $C$ and $T$ are

$$\log_{10}\mu = b_1 + \frac{b_2}{T - b_3} \tag{4a}$$

$$D = C \exp\left( c_1 + c_2 p_\infty - \frac{c_3 + c_4 p_\infty}{T} \right) \tag{4b}$$

where $b_1$, $b_2$, and $b_3$ are compositional vectors that depend on detailed chemical composition of the melt, including the contribution of $C$ via a multicomponent magmatic melt viscosity model[36]. $c_1 = -18.1$, $c_2 = 0.001888$, $c_3 = 9699$, and $c_4 = 3.626$ are coefficients valid when $p_\infty$ is given in GPa, and $T$ in K, following Zhang & Ni[37]. Eq. 4 is useful for modelling rhyolite melts, but for the peralkaline AOQ8 sample, Eq. 4a requires a different parameterization[38]

$$\begin{cases} b_1 = (-2.633 - 0.5727\ln\alpha) + (-0.2896 - 0.0002541\ln\alpha)C \\ \qquad\qquad b_2 = 11260 - 489.6\ln C \\ \qquad\qquad\qquad b_3 = 0 \end{cases} \tag{5}$$

where $\alpha$ is the mol.% oxides in the melt that are excess to charge-balancing roles. For AOQ8, a peralkaline solution to Eq. 4b is also required, which takes $c_1 = -16.55$, $c_2 = 0$, $c_3 = 10870$, and $c_4 = 1101$, after Zhang & Ni[37].

The bubble model uses the ideal gas law to relate the mass of gas in the bubble to volume and gas density. This model predicts $R(t)$ of a bubble in a shell. To upscale this to find the vesicularity $\phi$ of a magma parcel containing many bubbles, we define $\phi = V_g/(V_g + V_m)$ and $N_b = V_m^{-1}$, where $V_g$ and $V_m$ are the gas and melt volumes, respectively and $N_b$ is the number density of bubbles in the magma parcel (the number of bubbles per unit of melt volume). We then solve this bubble growth and resorption model under the assumption that $T$ and $p_\infty$ are the same for all $x$. At the gas–melt interface ($x = 0$), we impose the solubility condition $C = C_e$ (Eq. 1) given by $p_{H_2O}$ in the bubble and $T$. At the outer edge of the shell, we impose a zero-flux condition for $C$, and at all times $m$ is conserved. In this set up, we can apply an arbitrary time-varying $p$ and $T$ path, and output $\phi$.

## Model comparison with in-situ data

For the comparison with in-situ data in Fig. 3 we run the numerical model for 9 different bubble number densities $N_b$ and plot a weighted average $\phi(t)$ curve according to the measured effective $N_b$ distribution obtained from analysis of the tomography. Effective $N_b$ was estimated by determining the size distribution of vesicles in a tomograph and assuming that each vesicle is associated with a melt shell of appropriate volume $V_m$ to obtain the global vale of $\phi$. We chose a tomograph near the maximum measured $\phi$ value to minimize uncertainty associated with voxel resolution. The $\phi(t)$ curve produced by the modelling —particularly the maximum value of $\phi$ reached—is very sensitive to the initial concentration of $H_2O$ in the melt. We adjust this value to obtain a good visual fit to the data; the concentration values that we use

(0.087 wt% for the start of the up-ramp and 0.083 wt% for the start of the down-ramp) compare well with the measured one-atmosphere value (0.09 ± 0.03 wt%) for this melt composition. The model also includes a mass of argon in the starting bubble, reflecting the synthesis conditions. This mass is determined by computing the partial pressure of $H_2O$ at the glass transition temperature and adding sufficient argon to give the observed starting (pre-heating ramp) gas volume fraction.

## Application of the model to natural conditions

Idealised conditions (e.g., atmospheric pressure) and materials (e.g., AOQ8) are used in the experiments in order to isolate thermally driven growth and resorption, and to render the experiments tractable within the duration of a beamline experiment. When validating the model against the experimental data, the appropriate solubility, viscosity, and diffusivity laws for the AOQ8 composition are used. The theoretical and general nature of the modelling framework means that it, once validated, it can be used at other conditions and for other materials, if appropriate laws are used. Hence, when modelling natural rhyolitic obsidian, we use the component laws for viscosity, diffusivity, and solubility that are appropriate to that composition, and we use inputs to the model that are typical of eruptive rhyolites. These initial conditions are given and justified in the main text. We acknowledge that not all obsidian is homogeneous glass and that variations in $H_2O$, porosity, local crystallinity (including nanolites), and texture are common.

## Data availability

The data from the in-situ experiments in this study are available in the Supplementary Information.

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

## Acknowledgements

Funding was provided by Natural Environment Research Council grants NE/N002954/1 (E.W.L., M.C.S.H., J.P.C.), NE/X016668/1 (E.W.L., F.B.W., P.S.), NE/M018687 (K.J.D.), and NE/T00908X (K.J.D.), Engineering and Physical Sciences Research Council grant EP/T023198/1 (K.J.D.), and a beamtime access grant number EE16040 (E.W.L., F.B.W., K.J.D., M.C.S.H.) from the Diamond Light Source synchrotron.

## Author contributions

E.W.L., F.B.W., J.P.C., and M.C.S.H. conceived the study. E.W.L. and F.B.W. led the analysis and writing, with contributions from other authors. E.W.L., F.B.W., J.P.C., K.J.D., M.C.S.H., A.A., and T.C. undertook the beamline experiment that produced the tomography data. A.A., J.E.G., R.A.B., and M.N. created experimental materials. K.J.D., T.H., and C.A. segmented and analysed tomography data. P.S. and J.P.C. undertook numerical modelling.

## Competing interests

The authors declare no competing interests.
