## [Transparent Peer Review file · Nature Communications]

Obsidian forms by slow cooling

Corresponding Author: Professor Edward Llewellyn

Version 0:

Reviewer comments:

Reviewer #2

(Remarks to the Author)

Review of NCOMMS-25-68447-T "Obsidian forms by slow cooling" by Llewellyn, E.W., et al.

The manuscript argues that obsidian forms through slow cooling. Its central claim is supported by experimental and theoretical analyses of bubble resorption, indicating that vesicle-free obsidian must develop over much longer timescales than previously assumed. This study offers a compelling revision of prevailing views on obsidian genesis, and I recommend publication pending minor revisions.

1. Was the water content, and indeed the presence of water of the starting material confirmed in any way?
 2. Fig. 2 – might be worth it to explain in the figure caption what μ is as this manuscript is intended for broad audience, e.g., viscosity (μ). In the figures, particularly Fig. 4 a, and b I would suggest thicker lines and bigger font. In Fig. 4 a, and b I would also do something to distinguish the lines (maybe different colours like in the extended data? But with thicker lines) because it is a little hard to read.
 3. Lines 133-139: If there is enough space maybe it would be worth mentioning that this sluggish nucleation is due to kinetic effects (slow transport of the essential components) in this case.
 4. Materials and methods, lines 165-166. It is probably worth mentioning if the crucible was strictly Fe-free. Might also be worth it to include a table with the composition of the starting material (can be in the supplement).
- I am not fully convinced by the use of a video game, as an example in the manuscript but maybe it makes it more attractive, I will leave this to the authors to decide because it might be because of my complete lack of interest in video games.

Reviewer #3

(Remarks to the Author)

The authors challenge the long-held assumption that obsidian forms through rapid cooling of lava by using x-ray computed tomography of AOQ8, a lab-produced analog for obsidian, to show that to achieve a lack of gas bubbles/vesicles slow cooling and reabsorption is required. At magmatic temperatures the observe that bubbles shrink within the melt when cooled slowly. This is validated numerically, showing that bubble reabsorption is driven by increased H₂O solubility during cooling. While the claim is intriguing, I have concerns about the methodology and how the findings are generalized to all obsidian. They are as follow and should be addressed by the authors.

1. The experimental conditions are unrealistic. 1 atm of pressure is far below the pressures of silicic lavas, even at eruptive surfaces. H₂O solubility and mass transport is highly pressure dependent. Additionally, AOQ8 does not have the same properties as obsidian. Can the authors comment on why they did not take natural obsidian to elevated temperatures and perform cooling. Finally, can the authors comment on any potential issue with the use of Ar during the experiment, such as changes to surface tension or gas diffusion kinetics?
2. Comment on the differences in pressure used for the model (1 MPa) vs the experiment (1 atm).
3. The modeled resorption rates imply cooling timescales of days to months, which may be inconsistent with the known thermal histories of obsidian flows. The authors should clarify whether such slow cooling conditions are realistic for vesicle-free obsidian occurrences. Additionally, how does the lack of consideration for outgassing and composition affect the model?
4. The following scaling issue should be addressed and justified: Moving from synthetic glass with 1 vol.% vesicles to

natural magmas with 3–30 vol.% vesicles involves nonlinear diffusion and interface kinetics, but is presented with the same linear scaling.

5. The authors assume a slow-cooling homogenous resorption model, but many studies have shown that natural obsidian is rarely uniform, containing, for example, micro/nanolites and variable bubble densities. I suggest that the authors address the heterogeneity of natural obsidian as it relates to their work. Recent publications on this include: Bamber, et al. "3D quantification of nanolites using X-ray ptychography reveals syn-eruptive nanocrystallisation impacts magma rheology" (2025), which relates the presence of nanolites to change in rheology and eruptivity and Kennedy, et al. "Chemical and structural alterations in the amorphous structure of obsidian due to nanolites" (2022) which characterized nanolites in obsidian and their impact on local density and composition of amorphous material. Linking observed heterogeneity with your findings would be beneficial for generalizability of the results.

6. The conclusion that "Since all lavas must cool, this [bubbles resorb during slow cooling under specific conditions] is a general and global mechanism for obsidian formation" seems like an overgeneralization. Just because one plausible mechanism works for these experimental conditions does not mean it explains all obsidian formation.

7. Please be consistent with units of temperature - K or Celsius.

Version 1:

Reviewer comments:

Reviewer #2

(Remarks to the Author)

Thank you for addressing all my concerns. I really enjoyed this manuscript, and am excited to see it published.

Reviewer #3

(Remarks to the Author)

I had hoped that further discussion on heterogeneities in obsidian and how they align with your model would be included. I recognize that you have a sentence acknowledging them, but what the presence of crystallites, density differences, and chemical inhomogeneity in the nm and micron scale means in the context of your results would be valuable for understanding the formation of natural obsidian. Additionally, I still feel that the concluding sentences are an overgeneralization of the findings. That said, I appreciate the true novelty of your results and their implications for obsidian and glassy material cooling. I also appreciate the rigor and documentation of your methods. The authors have done enough to address my concerns and my own misunderstanding. For these reasons, I am fine with publication as is despite my preferences for minor revision.

Open Access This Peer Review File is licensed under a Creative Commons Attribution 4.0 International License, which permits use, sharing, adaptation, distribution and reproduction in any medium or format, as long as you give appropriate credit to the original author(s) and the source, provide a link to the Creative Commons license, and indicate if changes were

made.

Here we repeat the reviewer comments verbatim in grey, our replies to those comments in black, and we indicate any changes to the manuscript in **bold black**, for the editor's and reviewers' convenience. (Note that all changes in the manuscript are **highlighted**.) We have addressed every comment raised by the reviewers and in almost every case have made changes to the manuscript (where we have not made a change we have explained our rationale).

Reviewer #2

Review of NCOMMS-25-68447-T "Obsidian forms by slow cooling" by Llevellin, E.W., et al. The manuscript argues that obsidian forms through slow cooling. Its central claim is supported by experimental and theoretical analyses of bubble resorption, indicating that vesicle-free obsidian must develop over much longer timescales than previously assumed. This study offers a compelling revision of prevailing views on obsidian genesis, and I recommend publication pending minor revisions.

This is an accurate summary and we appreciate the positive comments.

1. Was the water content, and indeed the presence of water of the starting material confirmed in any way?

The initial water content resulting from the synthesis step was 0.09 ± 0.03 wt.%. This was confirmed using thermogravimetric analysis up to 1400 °C using a Netzsch Simultaneous Thermal Analyser at LMU, Munich, with 40.1 ± 0.5 mg of powder. This has now been added to the Materials and Methods in the manuscript. That addition is as follows:

Sintering synthesis was performed at 933 K for 7 hours yielding impermeable but bubbly glass with elevated concentrations of dissolved H₂O and argon-filled bubbles. **The concentration of H₂O in the glass was confirmed using a Netzsch Simultaneous Thermal Analyser (heating up to 1400 °C) to be 0.09 ± 0.03 wt.% H₂O.**

2. Fig. 2 – might be worth it to explain in the figure caption what μ is as this manuscript is intended for broad audience, e.g., viscosity (μ). In the figures, particularly Fig. 4 a, and b I would suggest thicker lines and bigger font. In Fig. 4 a, and b I would also do something to distinguish the lines (maybe different colours like in the extended data? But with thicker lines) because it is a little hard to read.

We now indicate the meaning of μ in the figure caption to Fig. 2.

We have made the lines in figure 4a and 4b 50% thicker. We prefer to avoid relying exclusively on colour to distinguish lines, particularly with a view to making the work accessible to those with colour vision deficiency (to this end **we have also added labels to the coloured lines in 4d**).

3. Lines 133-139: If there is enough space maybe it would be worth mentioning that this sluggish nucleation is due to kinetic effects (slow transport of the essential components) in this case.

This is a good suggestion from the reviewer. However, the sluggish nucleation of crystals is generally due to a combination of two effects: (1) the kinetics to which the reviewer refers, and (2) proximity to the liquidus, where supersaturations are low. Here, we do not distinguish these two effects and as such we prefer not to add a statement about kinetics, specifically. However, we do agree with the reviewer that kinetic lags can be one of the important factors.

4. Materials and methods, lines 165-166. It is probably worth mentioning if the crucible was strictly Fe-free. Might also be worth it to include a table with the composition of the starting material (can be in the supplement).

We now have added a confirmation that iron free crucibles were used. This addition is as follows.

Crystal-free samples of nominally anhydrous standard synthetic peralkaline rhyolite were produced using the AOQ system $A_{38}O_{34}Q_{28}$, where A, O, and Q refer to albite, orthoclase, and quartz, respectively. **In all synthesis steps, iron-free crucibles were used.**

We now include the glass composition in the Extended Data file as suggested.

I am not fully convinced by the use of a video game, as an example in the manuscript but maybe it makes it more attractive, I will leave this to the authors to decide because it might be because of my complete lack of interest in video games.

Obsidian plays a central role in the hugely popular video game Minecraft™, which has over 200 million active players around the world. It is probable, therefore, that Minecraft™ is the main reason that people – particularly young people – have heard of obsidian (that is certainly borne out by informal discussions with undergraduates studying Earth sciences). The game propagates the paradigm that obsidian is formed by rapid cooling/quenching (obsidian forms instantly when lava contacts water) and this is likely the most impactful source of that idea that high cooling rates are required to make obsidian. We believe there is a good chance that the paper's impact will go beyond the scientific community, exactly because we tap into this cultural reference to rapid cooling in the game Minecraft™. Therefore, we opt to retain this reference in the paper.

Reviewer #3

The authors challenge the long-held assumption that obsidian forms through rapid cooling of lava by using x-ray computed tomography of AOQ8, a lab-produced analog for obsidian, to show that to achieve a lack of gas bubbles/vesicles slow cooling and reabsorption is required. At magmatic temperatures the observe that bubbles shrink within the melt when cooled slowly. This is validated numerically, showing that bubble reabsorption is driven by increased H₂O solubility during

cooling. While the claim is intriguing, I have concerns about the methodology and how the findings are generalized to all obsidian. They are as follow and should be addressed by the authors.

This is a good summary of our manuscript. We thank the reviewer for their time.

1. The experimental conditions are unrealistic. 1 atm of pressure is far below the pressures of silicic lavas, even at eruptive surfaces. H₂O solubility and mass transport is highly pressure dependent. Additionally, AOQ8 does not have the same properties as obsidian. Can the authors comment on why they did not take natural obsidian to elevated temperatures and perform cooling. Finally, can the authors comment on any potential issue with the use of Ar during the experiment, such as changes to surface tension or gas diffusion kinetics?

We provide four main replies to these important points raised by the reviewer. They are:

1. ***Pressure conditions.*** Our core experimental goal was to demonstrate and quantify thermally driven bubble resorption, and to use those data to validate our numerical model. Since volatile solubility is also pressure dependent, it is essential that our experiments are conducted isobarically to eliminate pressure effects. There is currently no technology available that allows high-temperature beamline experiments to be conducted at elevated pressure *under strictly isobaric conditions*. This limitation does not apply to pressure-driven bubble growth, and the effectiveness of the same numerical framework when modelling bubble growth in natural rhyolite obsidian at elevated pressure has already been demonstrated in Coumans et al. (2021). It is therefore not feasible to conduct our experiments at elevated pressure, but it is also not necessary, since the pressure-dependence of solubility is already very well constrained (Liu et al., 2005) which allows us to account for the effect of pressure in the numerical model – i.e., the model can be used to investigate bubble growth and resorption at arbitrary pressures. We also note that the pressure *at* the eruptive surface is, by definition, one atmosphere for subaerial terrestrial eruptions, so the one-atmosphere experiments do have direct relevance to the natural context. We show results for low pressure resorption (at 1 MPa – i.e. approximately ten times atmospheric pressure) because many natural obsidians show evidence of very low H₂O (a compilation is provided in Wadsworth et al. 2020) that is consistent with the solubility at this low pressure (using the Liu et al. 2005 solubility model, which is calibrated for rhyolitic obsidian). This choice of low pressure is justified in the main text.
2. ***Natural obsidian composition.*** The low H₂O content of obsidian at one atmosphere would lead to impractically long bubble growth and resorption timescales for an in-situ beamline experiment using natural obsidian. AOQ and its variants (including the AOQ8 used here) are designed to be analogues of silicic melts in nature. For instance, it has been shown that idealised melt compositions such as AOQ reproduce the exact dependence of viscosity on temperature and H₂O content as seen in natural rhyolites of a wide range of compositions (see Hess & Dingwell 1996; note that therein the AOQ system is referred to as HPG). Crucially, the AOQ8 composition has the same viscosity and diffusivity behaviour as natural rhyolite, but shifted to lower temperatures, which allows thermally driven growth and resorption to occur on an experimentally viable timescale (i.e. a few hours, rather than months or years).
3. ***Relevance of experimental conditions.*** We use the experimental data to validate the model, then we use the model to investigate natural obsidian. The model is fully theoretical and general and so

once we have shown that it works well for the experiments, we can then be confident that it will work for the natural case as well. Crucially, when switching from validation of the experiments to application to nature, we switch from using a viscosity, solubility, and diffusivity laws for AOQ to using laws appropriate for natural rhyolitic obsidian. For these reasons, we do not believe that the idealised nature of the experimental materials or conditions is a weakness; instead, we claim it is a strength of our mixed experimental+theoretical approach.

4. **Issues with argon.** Parikh (1958) showed that the composition of the gas phase in contact with silicate melts has a negligible effect on surface tension (including testing argon). The value of surface tension in an argon atmosphere is within error of the value found for a variety of other atmosphere compositions. For this reason, we believe that argon does not affect the surface tension of silicate melts. There is no evidence that argon affects the diffusion kinetics of H₂O in melts, especially because the solubility of argon in melts is far lower than that of H₂O (Carroll & Stolper 1993).

We make the following additions to the manuscript to ensure that the content of these replies is embedded in the manuscript.

First, we explain the model application philosophy in the Materials and Methods in a new section as follows:

Application of the model to natural conditions. Idealised conditions (e.g. atmospheric pressure) and materials (e.g. AOQ8) are used in the experiments in order to isolate thermally driven growth and resorption, and to render the experiments tractable within the duration of a beamline experiment. When validating the model against the experimental data, the appropriate solubility, viscosity, and diffusivity laws for the AOQ8 composition are used. The theoretical and general nature of the modelling framework means that it, once validated, it can be used at other conditions and for other materials, if appropriate laws are used. Hence, when modelling natural rhyolitic obsidian, we use the component laws for viscosity, diffusivity, and solubility that are appropriate to that composition, and we use inputs to the model that are typical of eruptive rhyolites. These initial conditions are given and justified in the main text. We acknowledge that not all obsidian is homogeneous glass and that variations in H₂O, porosity, local crystallinity (including nanolites), and texture are common.

Second, in the Materials and Methods:

The use of argon as a carrier ensures that bubbles do not resorb to completion during post-sinter cooling. This is important for the subsequent beamline experiments because bubbles are available to grow as soon as the glass transition temperature is exceeded, without the need for stochastic and poorly constrained nucleation of bubbles. **Argon does not alter important material properties such as surface tension (Parikh 1958), and there is no evidence that this impacts viscosity or diffusivity of H₂O.**

2. Comment on the differences in pressure used for the model (1 MPa) vs the experiment (1 atm).

When modelling the experiments we use the pressure appropriate for the experiments (1 atm). Subsequently, when modelling the natural obsidian formation, we switch to using a different pressure in the model: 1 MPa in the main text, and we additionally report behaviour at 0.1 and 10 MPa in the extended data. This is a key advantage of using a physically based modelling framework – it can be applied to any conditions for which appropriate component models (e.g. for diffusivity, solubility, and viscosity) are available. The model is provided elsewhere (Coumans et al. 2020) and as such an interested reader could run it for any pressure-temperature-time pathway.

3. The modeled resorption rates imply cooling timescales of days to months, which may be inconsistent with the known thermal histories of obsidian flows. The authors should clarify whether such slow cooling conditions are realistic for vesicle-free obsidian occurrences. Additionally, how does the lack of consideration for outgassing and composition affect the model?

Our overall result is that cooling must be at least as slow as 10^{-5} - 10^{-7} °C/s. Other estimates of the cooling rate of obsidian-forming lavas include: (1) cooling rates inferred by spherulite growth of 10^{-5} °C/s (Yellowstone lavas; Befus et al. 2015); (2) 10^{-12} - 10^{-2} °C/s from geospeedometry of Banco Bonito lava (Kenderes & Whittington 2021); and (3) modelling of cooling of silicic lavas which indicates cooling can take up to decades (Manley 1992). There are other constraints, but these are some indicative values in reply to this comment. We have added to the main text as follows:

This calculation yields cooling rates in the range $10^{-7} \lesssim q_e \lesssim 10^{-5}$ °C/s, which corresponds to cooling over a month to a decade, **consistent with thermal modelling of rhyolite lavas (Manley 1992)**.

We do consider outgassing. We explore two scenarios: (1) obsidian is ultimately formed by collapse of an outgassing foam (our case where we use high initial porosities typical of foam collapse); and (2) obsidian is ultimately formed by sintering of pyroclasts which outgassed at the point of fragmentation (our case where we use a low initial porosity typical of sintering). In the submitted version we explicitly stated that the first scenario refers to “outgassed bubbly magma”. Now we make the same specification for the second scenario via the following change to the manuscript.

Respectively, these scenarios represent cooling of the final products of the collapsing foam model (outgassed bubbly magma) and of the cryptic fragmentation model (sintered products **outgassed at the point of fragmentation**);...

4. The following scaling issue should be addressed and justified: Moving from synthetic glass with 1 vol.% vesicles to natural magmas with 3–30 vol.% vesicles involves nonlinear diffusion and interface kinetics, but is presented with the same linear scaling.

We do solve explicitly for non-linear diffusion and interface kinetics in the model presented herein. There is no linear scaling involved.

5. The authors assume a slow-cooling homogenous resorption model, but many studies have shown that natural obsidian is rarely uniform, containing, for example, micro/nanolites and variable bubble densities. I suggest that the authors address the heterogeneity of natural obsidian as it relates to their work. Recent publications on this include: Bamber, et al. "3D quantification of nanolites using X-ray ptychography reveals syn-eruptive nanocrystallisation impacts magma rheology" (2025), which relates the presence of nanolites to change in rheology and eruptivity and Kennedy, et al. "Chemical and structural alterations in the amorphous structure of obsidian due to nanolites" (2022) which characterized nanolites in obsidian and their impact on local density and composition of amorphous material. Linking observed heterogeneity with your findings would be beneficial for generalizability of the results.

We now acknowledge that natural lavas are often heterogeneous as follows (in the Materials and Methods):

We acknowledge that not all obsidian is homogeneous glass and that variations in H₂O, porosity, local crystallinity (including nanolites), and texture are common.

6. The conclusion that "Since all lavas must cool, this [bubbles resorb during slow cooling under specific conditions] is a general and global mechanism for obsidian formation" seems like an overgeneralization. Just because one plausible mechanism works for these experimental conditions does not mean it explains all obsidian formation.

We respectfully disagree. The central point of this manuscript is that thermally driven resorption mechanism can form dense, bubble-free obsidian over wide-ranging conditions. *We do not only show that it is plausible under our experimental conditions*, on the contrary, we explore at length the extent to which thermally driven resorption applies in naturally relevant conditions (Figure 4 and Figures E1 – E6). The only other physical process that could account for the loss of all bubbles from rhyolitic melts during obsidian formation would be an increase in pressure; but this would not be consistent with the observation that many silicic lavas are vesicle-free regardless of the vertical position in the lava (see Tuffen et al. 2020). As our conclusion states: all lavas must cool. Hence, since the thermal resorption process can operate on naturally relevant cooling timescales, then that is a compelling argument that it occurs in all cases on Earth. Our conclusion does not claim that this is the only mechanism.

7. Please be consistent with units of temperature - K or Celsius.

We have now changed all references to temperature to Celsius throughout. Exceptions occur in the Materials and Methods where we introduce a model from a published reference and where that model is calibrated for inputs in kelvin.

References (used in this response but not included in the main text):

Befus, K.S., Watkins, J., Gardner, J.E., Richard, D., Befus, K.M., Miller, N.R. and Dingwell, D.B., 2015. Spherulites as in-situ recorders of thermal history in lava flows. *Geology*, 43(7), pp.647-650.

Carroll, M.R. and Stolper, E.M., 1993. Argon solubility in silicate melts and glasses: new experimental results for argon and the relationship between solubility and ionic porosity. *Geochim Cosmochim Acta*, 57, pp.5039-5051.

Kenderes, S.M. and Whittington, A.G., 2021. Faster geospeedometry: a Monte Carlo approach to relaxational geospeedometry for determining cooling rates of volcanic glasses. *Chemical Geology*, 581, p.120385.

Here we repeat the reviewer comments verbatim in grey, our replies to those comments in black, and we indicate any changes to the manuscript in **bold black**, for the editor's convenience. All changes in the manuscript are tracked (except for changing to numbered references).

The following response addresses the editorial steer: "Please add further discussion on the potential effects of heterogeneities and more nuance to your concluding remarks to aid our broad audience in interpreting your findings" by directly addressing Reviewer #3's comments.

Reviewer #3

I had hoped that further discussion on heterogeneities in obsidian and how they align with your model would be included.

We have now included a brief discussion of how heterogeneities fit within the conceptual model we propose. The following has been added to the discussion, citing a classic study on heterogeneous flow bands in obsidian:

Heterogeneities in natural obsidian often take the form of flow bands with differing (low) vesicularity [Gonnermann and Manga, 2005]; in our conceptual model these bands represent subtle differences in the initial vesicularity or initial H₂O content of the bands prior to cooling.

I still feel that the concluding sentences are an overgeneralization of the findings.

We have softened the final sentence.

Since all lavas must cool, we propose that this **provides** a general and global mechanism for the formation of obsidian.